# Demographic analysis of an Israeli *Carpobrotus* population

**Ana Bogdan**[1]*, **Sam C. Levin**[2,3], **Roberto Salguero-Gómez**[4], **Tiffany M. Knight**[2,3,5]

**1** Faculty of Biology and Geology, Department of Taxonomy and Ecology, Babeş-Bolyai University, Cluj-Napoca, Romania, **2** Institute of Geobotany, Martin Luther University Halle-Wittenberg, Halle (Saale), Germany, **3** German Centre for Integrative Biodiversity (iDiv) Halle-Jena-Leipzig, Leipzig, Germany, **4** Department of Zoology, University of Oxford, Oxford, United Kingdom, **5** Department of Community Ecology, Helmholtz Center for Environmental Research–UFZ, Halle (Saale), Germany

* anabogdann@yahoo.com

**Data Availability Statement:** All relevant data are within the manuscript and its Supporting Information files. The complete R code can be found at the following link https://github.com/

## Abstract

*Carpobrotus* species are harmful invaders to coastal areas throughout the world, particularly in Mediterranean habitats. Demographic models are ideally suited to identify and understand population processes and stages in the life cycle of the species that could be most effectively targeted with management. However, parameterizing these models has been limited by the difficulty in accessing the cliff-side locations where its populations are typically found, as well as accurately measuring the growth and spread of individuals, which form large, dense mats. This study uses small unmanned aerial vehicles (drones) to collect demographic data and parameterize an Integral Projection Model of an Israeli *Carpobrotus* population. We validated our data set with ground targets of known size. Through the analysis of asymptotic growth rates and population sensitivities and elasticities, we demonstrate that the population at the study site is demographically stable, and that reducing the survival and growth of the largest individuals would have the greatest effect on reducing overall population growth rate. Our results provide a first evaluation of the demography of *Carpobrotus*, a species of conservation and economic concern, and provide the first structured population model of a representative of the *Aizoaceae* family, thus contributing to our global knowledge on plant population dynamics. In addition, we demonstrate the advantages of using drones for collecting demographic data in understudied habitats such as coastal ecosystems.

## Introduction

Non-native invasive species are widely recognized as a major threat to native biodiversity, ecosystem function, and economic interests at a global scale [1,2]. Coastal ecosystems in particular provide vital ecosystem services and are threatened by biological invasions. Ecosystem services of coastal ecosystems include acting as a buffer against coastal hazards (wind erosion, tidal inundation during storm or hurricanes and wave overtopping), providing habitat for endemic plants and animals [3,4], and enabling recovery after a storm through succession processes [5]. Coastal ecosystems are prone to biological invasions, most notably by *Carpobrotus* spp., *Lampranthus* spp., *Opuntia* spp., *Malephora crocea* and *Mesembrianthemum cristallinum* [6,7].

levisc8/carpobrotus_IPMs/tree/master/Ana_Israel_IPM.

**Funding:** This work was funded by the Alexander von Humboldt Foundation under the framework of the Humboldt Professorship, by the Helmholtz Recruitment Initiative of the Helmholtz Association (both to TMK). SCL and TMK gratefully acknowledge the support of iDiv funded by the German Research Foundation (FZT 118 – 202548816). AB acknowledges the support of the Erasmus+ programme of the European Union to AB. RS-G was supported by a NERC Independent Research Fellowship (NE/M018458/1).

**Competing interests:** The authors have declared that no competing interests exist.

The succulent/subshrub species of the *Carpobrotus* N. Br. (*Aizoaceae*) genus are particularly problematic as they are "transformers" of sensitive coastal habitats. Most *Carpobrotus* species are native to the Cape region of South Africa, and some notorious species (e.g., *Carpobrotus edulis* (L.) N.E.Br., *Carpobrotus acinaciformis* (L.) L.Bolus) are now naturalized in Mediterranean climate regions on all continents except Antarctica [8]. *Carpobrotus* species decrease native plant and animal diversity in the places they invade [8–13]. *Carpobrotus* individuals compete directly with native plants for space, water, and nutrients [14,15], and indirectly by changing the soil pH [9], salt content, moisture level, nutrient content and microbial activity [8]. *Carpobrotus* litter disrupts natural soil nutrient cycles, increasing nitrogen and organic matter content and releasing allelochemicals that hinder seed germination, seedling emergence and root growth of some native plants [16–18]. *Carpobrotus* populations form mats that prevent sand movement and disrupt normal dune processes of disturbance and succession [9,19,20].

*Carpobrotus* disruptions to biotic and abiotic processes can hinder management efforts as they can persist for years after the invasion has been eradicated [21,22]. However, several restoration projects have successfully eradicated or reduced *Carpobrutus* populations and restored native plant and animal communities using either manual or chemical removal methods [13,23–25]. In Europe, conservation areas have dedicated ~1,000,000 EUR/year to controlling *Carpobrotus* invasions [8,26–28].

Structured population models (e.g., matrix population models [29], integral projection models [30]) of invasive plants provide a tool for generating comprehensive fitness estimates and identifying sensitive vital rates (e.g. survival, growth) to target with management [29,31–35]. Most research on *Carpobrotus* has focused on a determining its effects on other species in their ecological communities and on quantifying individual vital rates and their vulnerability to various management strategies [17,36–38]. To date, no studies have conducted a comprehensive demographic analysis of any species in the *Carpobrotus* genus. In addition, to our knowledge, stage-structured demographic studies on any member of the *Aizoaceae* family are non-existent, and most of the information we have on the population dynamics of succulents comes from the *Cactaceae* family [39].

While demography is a data-hungry and time-consuming enterprise in any habitat [40], demographic data are particularly challenging to collect in coastal populations, which can occur on inaccessible terrain, such as coastal bluffs [41]. Only a few prior studies have managed to quantify demographic variables safely and precisely in such challenging environments (e.g., by freehand climbing or by targeting individual plants that can be reached without climbing equipment [42–44]). The challenge of difficult terrain can be overcome by using unmanned aerial vehicles (UAVs), a rather recent technological advance in ecological monitoring studies [45–50]. UAVs allow sampling of coastal bluffs, which can otherwise be difficult and dangerous to reach [49,51]. UAVs provide high resolution imaging [46,48,49,52], and require little space for landing/take-off and minimal piloting expertise [46,47,51]. In addition, data collected from UAVs have been shown to be less error-prone than those collected directly by humans [53].

Here, we quantify the demography of a *Carpobrotus* population using small UAVs to collect data and construct IPMs. We project the asymptotic population growth rate ($\lambda$) and calculate sensitivities and elasticities of $\lambda$ to determine which vital rates (survival, growth, and reproductive success as both flowering probability and number of flowers) have the greatest effect on population growth rate if altered by management. We found that the size of *Carpobrotus* individuals predict its vital rates and that its demography is typical of a long-lived polycarpic plant. We demonstrate that drones allow for collection of demographic data in previously inaccessible locations, thus showing their great promise for filling global research gaps. We discuss the

consequences of our results for future *Carpobrotus* control actions and emphasize the need to target survival and growth of large plants.

## Methods

### Study site and genus

The study was conducted between April 2018 and April 2019 on a cliff top and cliff side overlooking the Mediterranean Sea, just north of Havatselet Ha'Sharon in Israel (32.364030 N, 34.857800 E, approx. elevation 25 m a.s.l., Fig 1). The area (hereby termed Havatselet) is characterized by heavy recreational use and is occasionally subject to small landslides from the cliff top down to the beach at the bottom. *Carpobrotus* is widespread in gardens in the surrounding area, and it is likely that this population escaped cultivation and established itself. We could not find any information on the age of invasion, but the size and consistent spatial distribution of the individuals suggests that it is a mature population.

Without genetic analysis, we were unable to determine exactly which species we worked with at Havatselet, but based on reports of species presence in Israel [20,54] it is safe to assume that most probably the species were *Carpobrotus edulis* (L.) N.E.Br., *Carpobrotus acinaciformis* (L.) L. Bolus, and *Carpobrotus chilensis* (Molina) N.E.Br., together with their hybrids, these being also the most problematic *Carpobrotus* species [16,55,56]. Hybrid *Carpobrotus* spp. can have unique behaviors in their invaded range, as hybridization can contribute to evolutionary changes in the dynamics of invasion [57]. Thus, our results may not apply to hybrids on other continents.

### Data collection

We developed a flight plan to map the population using DJI Ground Station Pro v2 (SZ DJI Technology Co.) for iPad (Apple Inc). This application allows users to generate polygons over

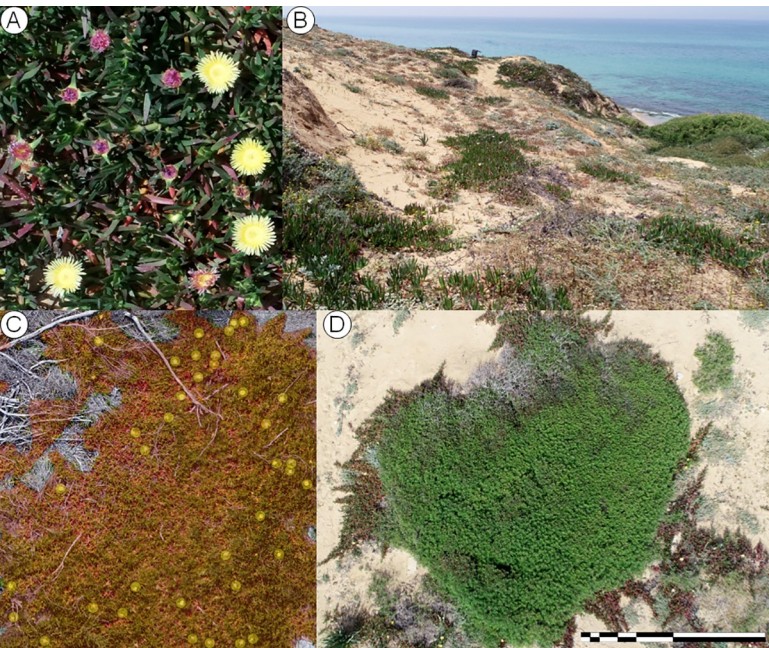

**Fig 1. Study site and *Carpobrotus* spp. ramets.** A close-up picture of the *Carpobrotus* spp. present at the Havatselet field site (a), a wider angle shot of part of the field site (b), polygons overlaid on a *Carpobrotus* spp. individual after orthomosaicing, as well as a point layer used to count flowers (c), and a single nadir image from the drone–generating the orthomosaic required calibrating 1695 and 1021 nadir images for 2018 and 2019, respectively (d).

areas of interest and computes optimal flight paths for surveying given a desired set of photo overlap, map resolution, and cameras on the drone itself. We selected a flight path that generated a resolution of ~0.35cm/pixel with ~85% side and front photo overlap, with the camera pointed 90 degrees downward (i.e. nadir imagery). The average width of open flowers is 7.22 cm (standard error: 0.042 cm) and unripe fruits was 2.2 cm (standard error: 0.014 cm) at the Havatselet population (SC Levin, unpublished data), so this resolution is sufficient to allow us to mark the majority of both in the resulting maps (orthomosaics, see below for more details). DJI Ground Station Pro only offers an option to fly at a specific altitude above the takeoff point, rather than keeping a fixed altitude over the terrain. Thus, the flight was conducted by hand so that we could maintain a consistent altitude over the variable terrain. Transects were flown with a DJI Phantom 4 Pro v1 (SZ DJI Technology Co.) along the lines of the flight plan generated by DJI GSP and images were recorded at 1.8 – 2m intervals. When batteries reached a critical level of power (i.e. 20%), we landed the drone, switched the batteries out, and the flight was resumed from the last stopping point. The estimated area covered was 2.063 hectares. The mission took approximately 1 h 40 min from 12:50 PM to 2.30 PM and five battery charges were necessary.

## Image processing and data extraction

Once all images were captured in 2018, individual photos were processed into a single composite, georeferenced orthomosaic map using Pix4Dmapper (Pix4D S.A., Switzerland). We drew individual polygons around contiguous sections of *Carpobrotus* plants (hereafter referred to as ramets), assigned each a unique ID number, and counted flowers using a point layer in QGIS 3.4.1 [58]. We repeated the process in 2019 by overlaying the 2019 map on the 2018 map and marking polygons for all surviving ramets as well as new recruits also in QGIS. Individual ramets that were not found again in 2019 were assumed dead. Maps were aligned between April 2018 and 2019 using the Georeferencer GDAL plugin (GDAL/OGR contributors 2020) in QGIS because the georeferencing of the orthomosaics was only accurate to ~5m. This plugin allows users to manually find and mark common points on both images and set them as reference points for transformation between the old coordinate system (map coordinates in 2019) and the new coordinate system (map coordinates in 2018). Pix4D generated multiple blocks of images during the orthomosaic optimization step in 2018, but not 2019, resulting in slight warping of the final image in 2018. We used the thin plate splines transformation for the coordinate systems and nearest neighbor resampling method. The reference points are available in the supplementary materials. In addition to drawing polygons around the individual ramets, we drew polygons around targets of known sizes and computed the ratio of calculated areas under the polygon vs the known sizes. We then re-scaled all computed ramet sizes using this ratio. Based on the ground truthing procedure, ramet sizes were underestimated by ~12% using the polygons from QGIS. Thus, all measures of size were re-computed before log transformation. The technique of overlaying maps from subsequent years adequately captured populations at resolutions high enough to identify ramets and flowers (data in S1 File). We identified 276 distinct ramets in 2018, re-found 233 in 2019, and were able to mark 10 new ramets in 2019. Georeferencing accuracy varied throughout the orthomosaic due to the blocks generated in 2018, but was never off by more than 4 m, and we are confident that we correctly identified each ramet in 2019 as either existing or new.

## Demographic modelling

To model vital rates, we considered how an individual's size ($z$) influences its survival, growth, and reproduction. We constructed a series of progressively more complicated generalized

**Table 1. Estimated regression coefficients of vital rates and their 95% confidence intervals.**

| Vital rate | Parameter | Estimated | Lower CI[a] | Upper CI[b] | Sample Size |
|---|---|---|---|---|---|
| Pr (Survival) | $\beta_{0,s}$ | 2.790 | 2.285 | 3.521 | 276 |
| | $\beta_{1,s}$ | 0.731 | 0.512 | 1.018 | |
| Growth | $\beta_{0,g}$ | 0.016 | -0.063 | 0.089 | 233 |
| | $\beta_{1,g}$ | 0.898 | 0.848 | 0.948 | |
| | $\beta_{\sigma_g}$ | -0.175 | -0.250 | -0.095 | |
| Pr (Flowering) | $\beta_{0,p_r}$ | 1.021 | 0.644 | 1.461 | 276 |
| | $\beta_{1,p_r}$ | 1.103 | 0.840 | 1.443 | |
| # of flowers | $\beta_{0,r_s}$ | 1.664 | 1.496 | 1.808 | 134 |
| | $\beta_{1,r_s}$ | 0.753 | 0.640 | 0.859 | |
| Recruit Size Distribution | $\mu_{r_d}$ | -3.103 | -3.690 | -2.521 | 10 |
| | $\sigma_{r_d}$ | 1.064 | 0.515 | 1.387 | 10 |
| Per-capita growth rate | $\lambda$ | 0.98 | 0.938 | 1 | |

[a,b]Confidence intervals were obtained by bootstrapping the data set 1000 times and re-fitting the model. Sample size indicates the number of ramets used in parameter estimation.

linear models for survival (*s(z)*) and growth to size *z'* conditional on survival (*g(z',z)*) using an intercept only, log-transformed surface area, and a 2nd-order polynomial of log-transformed surface area as an explanatory variable. Additionally, we fit generalized additive models with 8 knots and 6 knots for survival and growth, respectively, to determine graphically how well each model captured the mean trend [59]. The probability of reproducing (*p_r(z)*), and flower production (*r_s(z)*) had two competing models–the intercept-only, and the log-transformed surface area as fixed effects. The most parsimonious models for each vital rate were selected using AIC. Analysis of the residuals of the growth model showed a gradually declining variance as size increased, and so we fit a model where variance decreased exponentially with increasing size. As this model had a substantially lower AIC score than any other candidate model, we used it to predict both the mean and variance of the growth distribution. The growth model assumed errors followed a Gaussian distribution. Survival and probability of reproduction models were fit using binomial error distributions. The flower production model was first fit with a Poisson family, but analysis of the deviance showed overdispersion. Thus, we re-fit it with a quasi-Poisson to avoid underestimating standard errors for each coefficient. Results for the model fitting are shown in Table 1.

In addition to the regression models, we used our data to determine the size distribution of new recruits (*r_d(z')*) and the number of new recruits in 2019 per flower produced in 2018 (*r_r*). The recruit size distribution was modelled using a truncated normal distribution, and the rate at which flowers produced new ramets was modeled by dividing the number of new recruits in 2019 by the total number of flowers produced in 2018. *Carpobrotus* flowers take over a year to become mature fruits, and thus new recruits found in 2019 are the result of flowers produced prior to 2018. The most parsimonious model for *r_r* with our data considers flower production to be time invariant (i.e. the total number of flowers produced by the population is similar from year to year). We performed a regression parameter-based perturbation analysis to investigate how flower number variation affected our results [35]. We treated each parameter value in the flower number regression model as a Gaussian random variable and drew 1000 estimates from the distributions for each implied by the model estimation procedure. We then used parameter draw values for each parameter to rebuild the model 1000 times and computed

the per-capita growth rate again. The per-capita growth rate for the population is not very sensitive to these perturbations (S1 Fig), and so we feel comfortable that the underlying assumption did not substantially affect our results.

## Integral projection modelling and analysis

We combined the vital rate functions using an IPM [30,35] to estimate the population's asymptotic behavior. IPMs are similar to matrix projection models (MPMs; [29]) but allow for ramets to be classified by a mixture of discrete and continuous state variables, rather than discrete variables only as in MPMs. These models take the following general form:

$$n(z', t + 1) = \int_L^U K(z', z)n(z, t)dz$$

In this equation, *n(z', t + 1)* is a function describing the size structure of the population at time *t + 1*, *z'* is the state variable used to describe the population, and the integral of the expression over the domain of *z'* is the total population size. *L* is the lower limit of this domain, and *U* is the upper limit. *K(z',z)* is a bivariate kernel function that describes transitions to state *z'* given an individual's initial state, *z*, at time *t*. For this model, log-transformed surface area of ramets is the state variable *z*. *K(z',z)* is typically comprised of two or three sub-kernels that describe growth of existing ramets conditional on survival (*P*), sexual reproduction (*F*), and asexual reproduction (*C*). These sub-kernels are themselves comprised of functions describing the vital rates that contribute to them, and the functions are parameterized by the regression models described above. Because of *Caprobrotus*'s sprawling growth habit, we were unable to distinguish between growth of surviving ramets, and clone production when clones were contiguously part of the existing ramet (See Discussion). Thus, our model contained only two sub-kernels: *P* and *F*. Substituting these into Eq 1, our model took the form:

$$n(z', t + 1) = \int_L^U [P(z', z) + F(z', z)]n(z, t)dz$$

Analytical solutions to these integrals are not available, and so a numerical solution is necessary [35,60]. We solved the equation numerically using the midpoint rule of integration with 100 meshpoints along the domain [*L*, *U*] to generate a 100 by 100 iteration matrix, *K*. We checked whether using different numbers of meshpoints affected our results by rebuilding the model from 100 to 500 meshpoints at increments of 50 meshpoints each time. The change in per-capita growth rate was on the order of $10^{-5}$, so we retained the model with 100 meshpoints and continued with the analysis. A complete description of the model functions is in the data of the S1 File.

Once the IPM was implemented, we then calculated the per-capita growth rate as the single time step growth rate after iterating the model to reach asymptotic dynamics ($\lambda$) [35]. We also computed the sensitivity and elasticity functions of $\lambda$ at the kernel and sub-kernel level to understand the relative contributions of survival/growth transitions and reproductive transitions to the population growth rate [35]. To quantify uncertainty in our data, we resampled our demographic data with replacement 1,000 times and refit all vital rates using the same functional forms. We constructed IPMs for each iteration and computed per-capita growth rates, sensitivities, and elasticities. Thus, all estimates reported here include our point estimate and the 95% confidence intervals from the bootstrapping procedure.

## Results

Ramets ranged in size from -6.04 to 3.31 m$^2$ (natural log scale), and flower counts were heavily skewed towards smaller values with a few very large, highly reproductive ramets (Fig 2). Vital

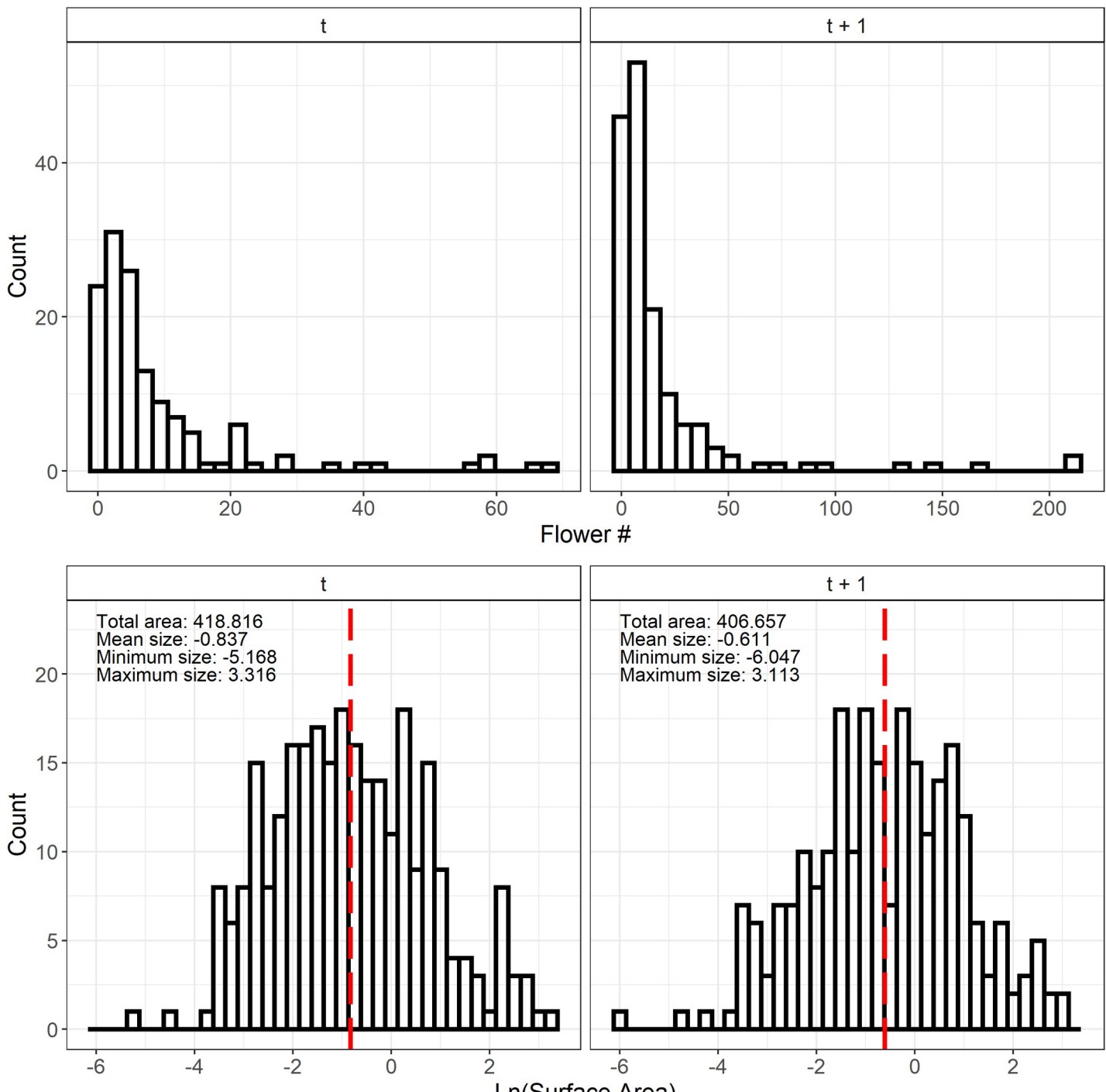

**Fig 2. Frequency distributions for flowers and ramets.** Top row: Frequency distributions of the number of flowers at time *t* and *t+1*. Bottom row: Frequency distributions of ramet size at *t* and *t+1*. Red dashed vertical lines denote the mean ramet size for each sampling time, and minimum and maximum sizes are indicated in the top left corner.

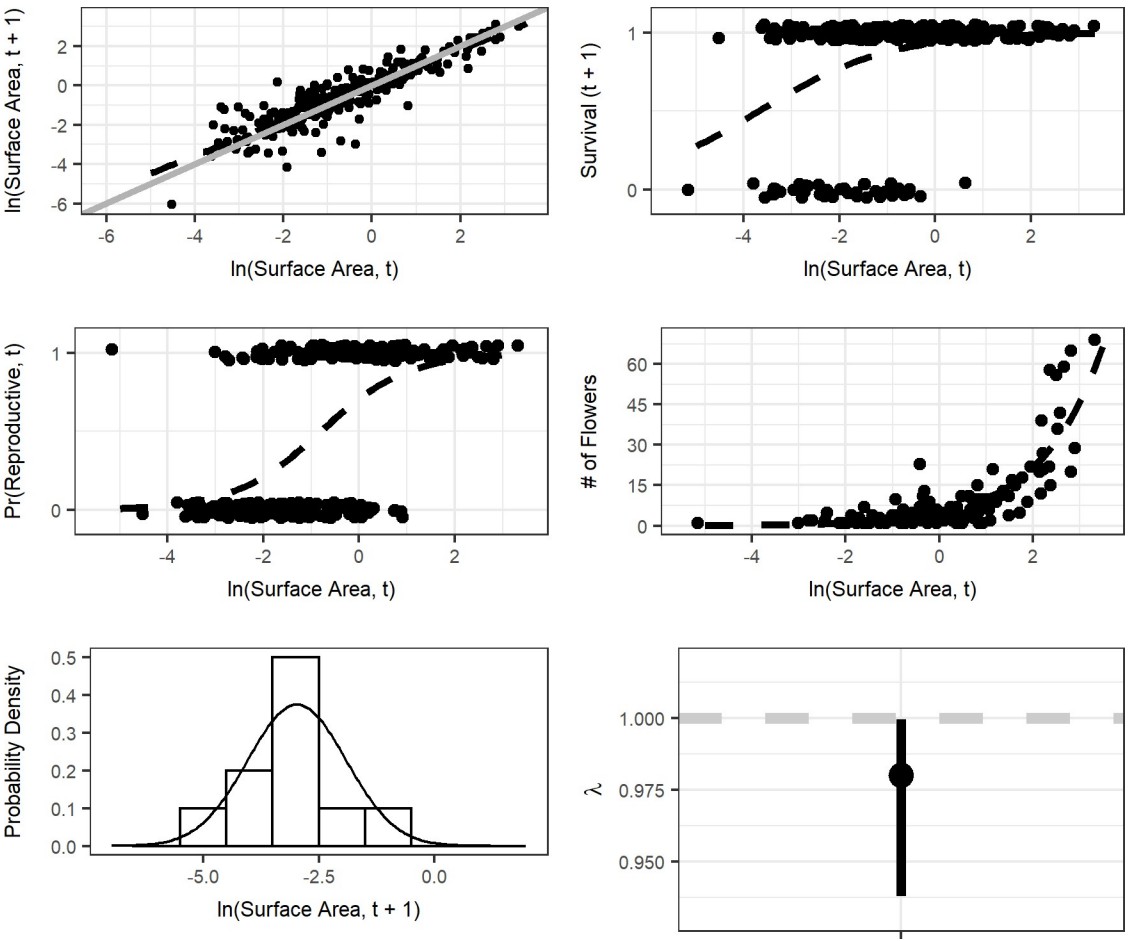

**Fig 3. Vital rates models.** Results for ramet growth (a), survival (b), probability of flowering (c), flower production (d), and recruit size distribution (e), and per-capita population growth rate with 95% bootstrap confidence intervals (f). The dotted black lines in a-c, e and the solid black line in d are fitted relationships from the models described in Methods. In (a), the solid grey lines show a 1:1 relationship (i.e. no change in ramet size), and the dotted grey line in f denotes a per-capita population growth rate of 1 (e.g. population level stasis).

rate regression using log-transformed surface area of ramets of *Carpobrotus* spp. as a fixed effect yielded good fits for all of our vital rates and generally lower AIC scores than intercept-only, quadratic fits, or GAMS (Fig 3, Table 1, data in S1 File). GAMs and polynomial fits did not substantially improve the overall model fit or substantially lower the AIC score for each vital rate, even with the increased complexity of the model itself. Thus, our IPM only includes models with a single fixed effect–log-transformed surface area–as explanatory variables (data in S1 File).

Our estimate of the population growth rate was $\lambda = 0.98$ (Fig 3, Lower 95% CI = 0.938, Upper 95% CI = 1). These estimates indicate that the population would not significantly depart from demographic stability on the long-term under a constant environment and density independence (Fig 3). The IPM's sensitivity surface indicates population growth rates are most sensitive to small ramets growing large rapidly, and the elasticity analysis indicates that survival/growth transitions are far more important to $\lambda$ than reproduction (Fig 4).

## Discussion

Our results show a *Carpobrotus* population with a stable growth rate (Table 1, Fig 3), indicating that this population is neither declining nor expanding if the environmental conditions

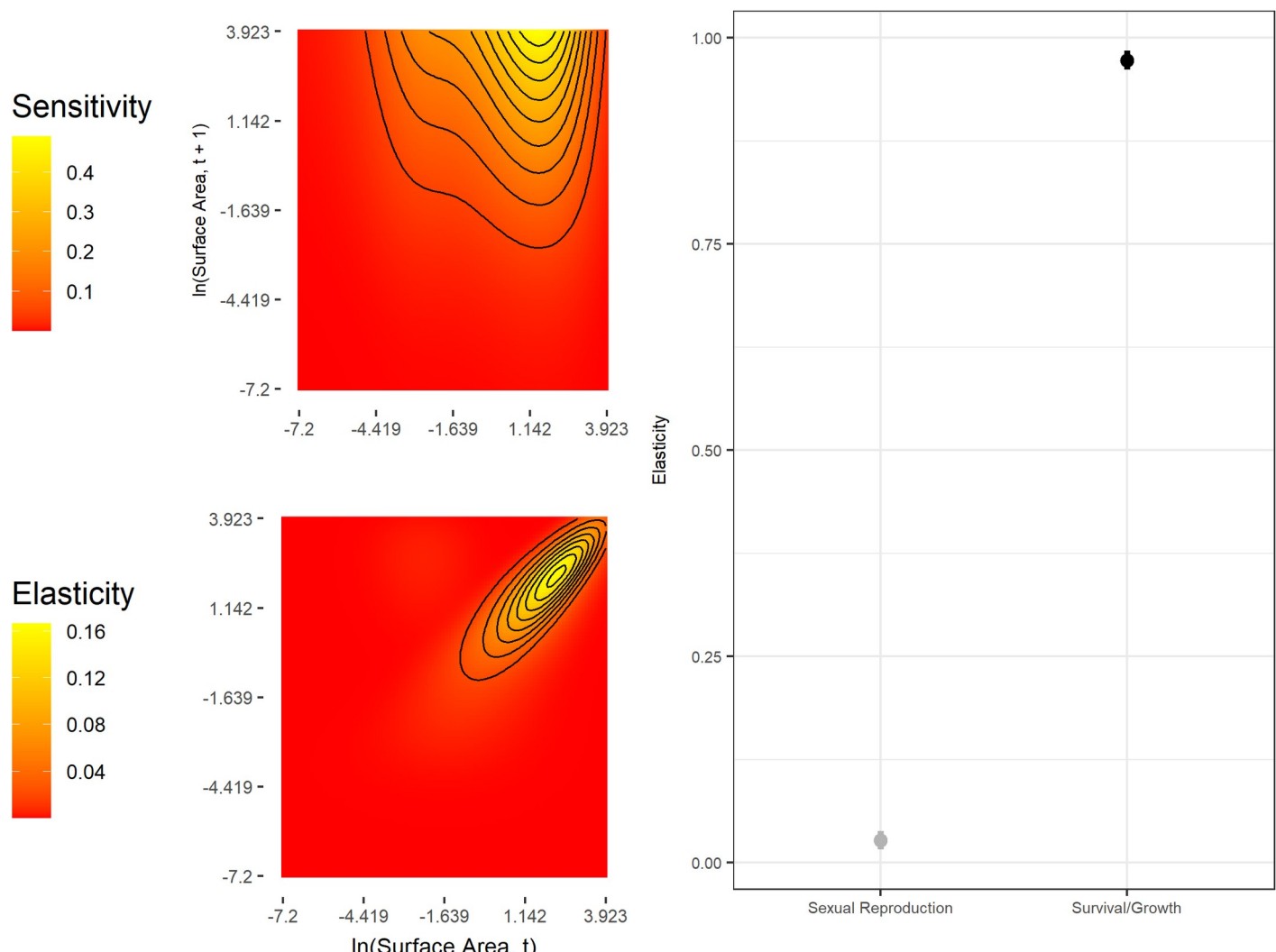

**Fig 4. Sensitivity and elasticity functions of λ.** Left: Sensitivity and elasticity kernels for the complete model. Right: Elasticities for each sub-kernel, indicating relative contributions of each to the per-capita growth rate. The range line for each point estimate of the sub-kernel level elasticity shows the 95% confidence interval derived from the boot strapping procedure.

under which it was examined do not change. There are a few reasons that might explain why *Carpobrotus*, which is known to be highly invasive in other places [8,36–38], might not be expanding at our site in Israel. First, our Havatselet population might be an old population that has reached an 'equilibrium' and is now limited by space [34,61,62]. The population grows on a cliff side and may be difficult for it to spread to unoccupied locations. Second, the Israeli climate might not be ideal for *Carpobrotus*. The Cape Region, where these plants are native, has a cold semi-arid climate [63], with average monthly temperatures reaching 21.60˚C [8]. Our Havatselet site has a warm Mediterranean climate [63] with higher temperatures, reaching a mean of 27.0˚C. Summers in Israel are much drier, often with no precipitation, compared to the minimum of 30 mm of rain the summer months of Cape Region receives [64]. Third, it is possible that stable population growth rates are typical for older, less disturbed *Carpobrotus* populations. Rapid ramet and population growth rates may characterize earlier stages of invasion, and slow down occurs once a site becomes more saturated. Thus, newer or

more recently disturbed populations are likely to exhibit different behavior from the Havatse-let population. Finally, we note that our study is one year long, and that environmental conditions across time, especially in the context of climate change, which might increase or decrease the population growth rate.

We find that the growth rate of our *Carpobrotus* population is highly sensitive to changes in survival and growth of ramets and less sensitive to reproduction (Fig 4). Specifically, the sensitivity analysis indicates that rapid growth of ramets that are already at a moderate size would have dramatic effects on $\lambda$. We did not observe any ramets in our study having such dramatic changes in size and it might not be biologically possible for *Carpobrotus*. Our result that elasticity of $\lambda$ is high for survival and low for sexual reproduction is expected. Across the plant and animal kingdoms, elasticity of $\lambda$ to survival is known to increase with life span [34,65–68], suggesting that persistence of healthy adults is critical to maintaining populations of long-lived perennial plants.

Eradication of *Carpobrotus* is the objective of many coastal management programs. Our elasticity analysis shows the importance of targeting large ramets for removal and our methods show the possibility of using drones to rapidly monitor populations and allow for adaptive management [69]. A variety of tactics have successfully killed *Carpobrotus* and allowed for native biodiversity recovery. Manual removal is feasible for accessible populations, and it may or may not be optimal to follow this up with the removal of dead shoots and litter. Leaving litter increases the risk *Carpobrotus* germination from seed and slows recovery of native plants, but decreases the problem of erosion, which is an important issue in sites with steep slopes [13,23,25]. Effective mortality of *Carpobrotus* and rapid recovery of native vegetation has also been demonstrated with chemical control using Glyphosate [18,24]. At our Havatselet site, both of these methods would be challenging due to the presence of large ramets on the face of the cliff. Biological control agents that reduce survivorship are under consideration, such as the soft-scale insect *Pulvinariella mesembryanthemi* and the generalist pathogenic fungus *Sclerotinia sclerotiorum* [8]. Regardless of the control method used, rapid evaluation of the population size and structure would help any management program, as it would provide the information needed for the learning process in adaptive management [69]. If the post-data processing procedures could be better optimized, the drone technology has great potential to allow for rapid monitoring of *Carpobrotus* populations.

We suggest, as other researchers have [8,54,70–72], that prevention is critical to curb regional scale invasion (i.e. landscape or political entity level) of *Carpobrotus*. To our knowledge, Israeli legislation does not include rules for the introduction and control of *C. edulis* and *C. acinaciformis*, and as such these are still allowed to be planted in urban gardens. This greatly hinders efforts for invasion prevention. Further demography studies could contribute to the identification of high-risk areas for which regulations could be put in place [8,54,73].

We were successful in using the drone for creating the maps of the *Carpobrotus* population and later use those maps for evaluating vital rates and parameterizing an IPM. This is another potential ecological use case for a rapidly developing technology used in both animal [53,74,75] and plant population studies [49,76,77]. While in our case, the drone successfully generated maps and subsequent estimates of ramet size, we acknowledge that they have their limitations. Georeferencing can prove challenging in environments without many permanent features to mark across multiple samples. If maps are not well aligned, re-identifying individuals from the previous year can be quite challenging. Advances in RTK and PPK GPS technology have greatly reduced this challenge, though these can still be quite pricey relative to typical conservation budgets. Furthermore, size estimates can vary greatly when the absolute distance between the ground and the camera varies across the orthomosaic scene, necessitating the placement of ground truth targets in areas that may be difficult to physically reach.

Additionally, dense vegetation mats can confound the orthomosaic generating procedure. Even when it is successful, it may be difficult to identify individual plants. Therefore, we recommend that vegetation density and height be considered when choosing the mapping method.

In conclusion, we confirm that the size of different *Carpobrotus* ramets is a reliable predictor for vital rates, and that the demography of *Carpobrotus* is typical of a long-lived polycarpic plant. We demonstrated that drones can successfully allow for demographic data collection for plant species that were previously inaccessible to researchers. Thus, using this technology in future demography research shows great promise for filling global data gaps. Our results emphasize the need to target survival and growth of large ramets in control programs.

## Supporting information

**S1 Fig. Results for the flower number regression parameter perturbation analysis.** The red point is the per-capita growth rate estimated from the regression parameters. The black line shows the upper and lower 95% confidence intervals derived from the draws that perturbed each flower number regression parameter. The range of values is quite small, and our qualitative results are not affected by the assumption of little temporal variation in flower production. (TIF)

**S1 File. IPM functions, drone results and model fitting results. S1.1 Table. Statistical values for the vital rates parameters.** Coefficients, standard errors, test statistics, and p-values for each of the vital rate regression used in the IPM to generate point estimates of lambda, sensitivity, and elasticity functions. **S1.2 Table.** AIC model selection tables for candidate growth models. S1.3 Table. AIC model selection tables for candidate survival models. S1.4 Table. AIC model selection tables for candidate probability of flowering models. S1.5 Table. AIC model selection tables for candidate flower production models. (PDF)

## Acknowledgments

We thank the two anonymous reviewers and the editor for their insightful suggestions which improved the previous version of this work.

## Author Contributions

**Conceptualization:** Ana Bogdan, Sam C. Levin, Roberto Salguero-Gómez, Tiffany M. Knight.

**Data curation:** Sam C. Levin.

**Formal analysis:** Ana Bogdan.

**Funding acquisition:** Tiffany M. Knight.

**Methodology:** Ana Bogdan, Sam C. Levin.

**Software:** Sam C. Levin.

**Supervision:** Roberto Salguero-Gómez, Tiffany M. Knight.

**Validation:** Sam C. Levin.

**Writing – original draft:** Ana Bogdan, Sam C. Levin.

**Writing – review & editing:** Sam C. Levin, Roberto Salguero-Gómez, Tiffany M. Knight.

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
