## [Decision Letter · Decision Letter 0]

24 Feb 2021

PONE-D-20-29490

Demographic analysis of Israeli Carpobrotus populations: management strategies and future directions

PLOS ONE

Dear Dr. Bogdan,

Thank you for submitting your manuscript to PLOS ONE. After careful consideration, we feel that it has merit but does not fully meet PLOS ONE’s publication criteria as it currently stands. Therefore, we invite you to submit a revised version of the manuscript that addresses the points raised during the review process.

We look forward to receiving your revised manuscript.

Kind regards,

Sergio R. Roiloa

Academic Editor

PLOS ONE

Journal Requirements:

Additional Editor Comments (if provided):

We have received the reports of two reviewers. Both consider that the document includes interesting and novel information with potential for publication in PLOSONE, and I agree. Please, incorporate/respond the changes/comments raised by the reviewers.

Reviewers' comments:

Reviewer's Responses to Questions

**Comments to the Author**

1. Is the manuscript technically sound, and do the data support the conclusions?

Reviewer #1: Yes

Reviewer #2: Partly

2. Has the statistical analysis been performed appropriately and rigorously? 

Reviewer #1: Yes

Reviewer #2: I Don't Know

3. Have the authors made all data underlying the findings in their manuscript fully available?

Reviewer #1: Yes

Reviewer #2: Yes

4. Is the manuscript presented in an intelligible fashion and written in standard English?

Reviewer #1: Yes

Reviewer #2: Yes

5. Review Comments to the Author

Reviewer #1: This is a very interesting paper about an invasive plant secies (Carpobrotus sp. plur.) in a Mediterranean site. This topic still poor investigated, at least in Western Mediterranean. Analyses are interesting performing a strong statistic. Text is well written. I would like to read some things about control and management of these populations, also adopting conservation project cycle (see Hockings et al., 2006, IUCN). A sentence usful to managers could be useful (see below in suggestions). However, I think the this ms could deserves to be published on PlosONE after MINOR REVISIONS. I have only minor suggestions and comments. I reported them below, hoping that could improve a bit the first draft of the manuscript.

In the next number of Folia Geobotanica there will be a paper of Battisti & Fanelli on the dispersal of Carpobrotus is a Mediterranean island. I suggest to read it and cite both this paper and the references therein (a large review of this topic has been reported).

r. 62. I think that also Malephora crocea and Mesembrianthemum cristallinum should be added as invasive plants along rocky Mediterranean coasts. See Ecology, Ethology and Evolution Battisti & Fanelli n press.

r. 85 For population analysis of dunal plants see also Garzia et al., 2019. Aliens come from the edge: a distribution pattern of focal alien plants in a small coastal reserve. Quaderni del Museo Civico di Storia Naturale di Ferrara, 7: 113-119, ISSN 2283-6918. Available on: https://storianaturale.comune.fe.it/modules/core/lib/d.php?c=BESU4

r. 107, 117 and everywhere: ‘spp.’ should be written not in italic.

r. 235. Why only edulis has been reported?

row 330. ‘as other researchers have’ Who? Add references.

I am a wildlife manager and in nature reserve who I manage there are a large number of nuclei of this plant. I noted as literature about problem-solving and project management techniques aimed to control these populations is very scanty. There are a lage number of research about phenology, ecology, dispersal, competition but very few papers about operational and pragmatic techniques useful to managers. This is a sort of ‘paralys by analysis’ (i.e focus only on data sampling and not on operational control of this species), a problem yet reported in conservation biology. I would read some sentences in thsi regard (as ‘suggestions for managers’).

References 51, 52, 54: These websites should be cited in this way?

Check further for references and citations in the text.

Add the role of anonymous reviewers and Editors in the acknowledgments.

Have a nice work.

Reviewer #2: General comments

I revised the manuscript entitled "Demographic analysis of Israeli Carpobrotus populations: management strategies and future directions".

I found some merits in the manuscript which deals an interesting topic, providing demographic data using drones. Nevertheless, the whole paper should be extensively revised and rewritten.

Title:

The title should be revised and focused on the type of research, the use of images obtained by drone. The study has only been conducted in one population. “Management strategies and future directions” do not seem justified to include it in the title.

Abstract:

Line 22 (L22) delete (UAVs).

L29 – Aizoaceae italics.

L31 – confirm by demonstrate.

Introduction

The introduction should be restructured. Information is there but is pretty messy.

Invasive species and the importance of coastal ecosystems (L66-72) could be reduced and could join with the problem of invasive species (L35-36).

The description of Carpobrotus (L37-43), its impacts (L44-51) and the investigations (L56-59) should be rewritten and joined in a paragraph. Research on eradication and the effects of control actions should be mentioned. See below recent papers:

Chenot et al. (2018) Restor Ecol 26:106-113.

Lazzaro et al. (2020) Biologia 75:199-208.

Buisson et al. (2020), Applied Vegetation ScienceVolume 24, Issue 1.

Braschi et al. (2021) Biodiversity and Conservation (2021) 30:497–518.

Fos et al. (2021) Plant Biosystems.

The part dedicated to "Structured population models" in Carpobrotus (L 52-59, L 62- 65) and "Structured population models" in dune ecosystems (L72-79) should be shortened and joined in one paragraph.

L37 - Carpobrotus N. Br. (Aizoaceae).

L38 – sensitive by frail.

L40 – Correct botanic names C. edulis and C acinaciformis.

L50 – Allelopathic properties of the Carpobrotus litter also inhibit emergence (Fos et al., 2021.

L55 – can be rewritten: “management, managers, management” in the same sentence.

L59-62 – “Succulents are an …. and Opuntia spp (29)” delete the paragraph, the information is not relevant.

L86-87 delete “unmanned aerial vehicles” and “Integral Projection Models”.

L92 – confirm by demonstrate.

Methods

The methods are written in a single section and they must organize in specific sections.

Fig. 1 - include scale.

L112-119 reduce paragraph, relevant information is only L115-116.

The use of the terms “plant” and “ramet” is confusing.

L160 4 m instead of 4m.

L190 delete “Integral Projection Model”.

Results

The results should include descriptive information from the experimental data, for example: total area occupied by Carpobrotus, mean plant area, maximum and minimum plant area, frequency by size of plant area, number of flowers and fruits per plant, etc. and variations from 2018 to 2019.

L 235 why C. edulis ???.

Discussion

The discussion should be revised, shortened and rewritten.

Include table and figure references in the discussion.

L273 – Aizoaceae italics.

L279 - delete “Integral Projection Models”.

L269-275 - This paragraph is better included in the introduction.

L282-284 - Delete “Individual …… vertical cliff”.

L289-291 “Third, it is … for Carpobrotus” - this sentence is contradictory with the references on invasive behavior and growth of Carpobrotus, (see Campoy et al 2018),

L297-298 - No data is presented regarding it, see Result comments.

L303-329 - This part should be rewritten by joining the 2 paragraphs. Recent eradication works and the effects of control actions should be included and discussed. See papers in introduction comments. It would be interesting to discuss the contributions of this work and the potential of the use of drones at least in the monitoring of regenerated areas.

L305-306 – “It is … growth” too speculative.

L310-311 - In the case of Carpobroptus, the actions should be aimed at the complete eradication and not at the reduction of the population.

L330-338 - This part is very general and it has nothing to do with the work they present.

6. PLOS authors have the option to publish the peer review history of their article (what does this mean?). If published, this will include your full peer review and any attached files.

Reviewer #1: No

Reviewer #2: No

---

## [Author Response · Author response to Decision Letter 0]

9 Apr 2021

Reviewer #1: 

This is a very interesting paper about an invasive plant secies (Carpobrotus sp. plur.) in a Mediterranean site. This topic still poor investigated, at least in Western Mediterranean. Analyses are interesting performing a strong statistic. Text is well written. I would like to read some things about control and management of these populations, also adopting conservation project cycle (see Hockings et al., 2006, IUCN). A sentence usful to managers could be useful (see below in suggestions). However, I think the this ms could deserves to be published on PlosONE after MINOR REVISIONS. I have only minor suggestions and comments. I reported them below, hoping that could improve a bit the first draft of the manuscript.

We thank Reviewer#1 for his/her nice evaluation. In addition to expanding our Discussion section to include more about the control and management of Carpobrotus spp. populations, we also incorporate the Hockings et al. 2006 paper into our cited literature. See lines 292-308, which now read:

Eradication of Carpobrotus is the objective of many coastal management programs. Our elasticity analysis shows the importance of targeting large ramets for removal and our methods show the possibility of using drones to rapidly monitor populations and allow for adaptive management [69]. A variety of tactics have successfully killed Carpobrotus and allowed for native biodiversity recovery. Manual removal is feasible for accessible populations, and it may or may not be optimal to follow this up with the removal of dead shoots and litter. Leaving litter increases the risk Carpobrotus germination from seed and slows recovery of native plants, but decreases the problem of erosion, which is an important issue in sites with steep slopes [13,23,25]. Effective mortality of Carpobrotus and rapid recovery of native vegetation has also been demonstrated with chemical control using Glyphosate [18,24]. At our Havatselet site, both of these methods would be challenging due to the presence of large ramets on the face of the cliff. Biological control agents that reduce survivorship are under consideration, such as the soft-scale insect Pulvinariella mesembryanthemi and the generalist pathogenic fungus Sclerotinia sclerotiorum [8]. Regardless of the control method used, rapid evaluation of the population size and structure would help any management program, as it would provide the information needed for the learning process in adaptive management [69]. If the post-data processing procedures could be better optimized, the drone technology has great potential to allow for rapid monitoring of Carpobrotus populations.

In the next number of Folia Geobotanica there will be a paper of Battisti & Fanelli on the dispersal of Carpobrotus is a Mediterranean island. I suggest reading it and cite both this paper and the references therein (a large review of this topic has been reported).

At the time in which we completed this revision, the next issue of Folia Geobotanica was still not available, and we have not been able to find this work available in some online archive, such as bioRxiv or similar. 

r. 62. I think that also Malephora crocea and Mesembrianthemum cristallinum should be added as invasive plants along rocky Mediterranean coasts. See Ecology, Ethology and Evolution Battisti & Fanelli n press. 

We have included the suggested invasive species and cited the relevant paper in the Introduction section. See lines 43-44, which now read: Coastal ecosystems are prone to biological invasions, most notably by Carpobrotus spp., Lampranthus spp., Opuntia spp., Malephora crocea and Mesembrianthemum cristallinum [6,7].

r. 85 For population analysis of dunal plants see also Garzia et al., 2019. Aliens come from the edge: a distribution pattern of focal alien plants in a small coastal reserve. Quaderni del Museo Civico di Storia Naturale di Ferrara, 7: 113-119, ISSN 2283-6918. Available on: https://storianaturale.comune.fe.it/modules/core/lib/d.php?c=BESU4

We thank Reviewer #1 for letting us know about these recent and upcoming papers. We have added the information from these papers and their citations. See lines 75-78 which now read:

Only a few prior studies have managed to quantify demographic variables safely and precisely in such challenging environments (e.g., by freehand climbing or by targeting individual plants that can be reached without climbing equipment [42-44]). 

r. 107, 117 and everywhere: ‘spp.’ should be written not in italic.

We have made the suggested change throughout the manuscript. 

r. 235. Why only edulis has been reported?

We have revised the phrase, and changed that particular portion of the Results section to Carpobrotus spp., as all the listed species (Carpobrotus edulis (L.) N.E.Br., Carpobrotus acinaciformis (L.) L.Bolus and Carpobrotus chilensis (Molina) N.E.Br.) were included in our analysis.

row 330. ‘as other researchers have’ Who? Add references. 

We have added the following references:

6. Campoy JG, Acosta ATR, Affre L, Barreiro R, Brundu G, Buisson E, et al. Monographs of invasive plants in Europe: Carpobrotus. Bot Lett. 2018;165(3–4):440–75.

48. Dufour-Dror J-M. Israel’s Least Wanted Alien Ornamental Plant Species. 2013;1-21.

76. Leung B, Lodge DM, Finnoff D, Shogren JF, Lewis MA, Lamberti G. An ounce of prevention or a pound of cure: bioeconomic risk analysis of invasive species. Proc R Soc London Ser B Biol Sci. 2002;269(1508):2407–13. 

77. César de Sá N, Marchante H, Marchante E, Cabral JA, Honrado JP, Vicente JR. Can citizen science data guide the surveillance of invasive plants? A model-based test with Acacia trees in Portugal. Biol Invasions. 2019;21(6):2127–41. 

78. Hulme PE. Plant invasions in New Zealand: global lessons in prevention, eradication and control. Biol Invasions. 2020;22(5):1539–62. 

I am a wildlife manager and in nature reserve who I manage there are a large number of nuclei of this plant. I noted as literature about problem-solving and project management techniques aimed to control these populations is very scanty. There are a lage number of research about phenology, ecology, dispersal, competition but very few papers about operational and pragmatic techniques useful to managers. This is a sort of ‘paralys by analysis’ (i.e focus only on data sampling and not on operational control of this species), a problem yet reported in conservation biology. I would read some sentences in thsi regard (as ‘suggestions for managers’).

We have expanded our discussion to add more of the relevant literature on suggestions for managers, and how this also relates to our findings. See lines 292-308, which now read:

Eradication of Carpobrotus is the objective of many coastal management programs. Our elasticity analysis shows the importance of targeting large ramets for removal and our methods show the possibility of using drones to rapidly monitor populations and allow for adaptive management [69]. A variety of tactics have successfully killed Carpobrotus and allowed for native biodiversity recovery. Manual removal is feasible for accessible populations, and it may or may not be optimal to follow this up with the removal of dead shoots and litter. Leaving litter increases the risk Carpobrotus germination from seed and slows recovery of native plants, but decreases the problem of erosion, which is an important issue in sites with steep slopes [13,23,25]. Effective mortality of Carpobrotus and rapid recovery of native vegetation has also been demonstrated with chemical control using Glyphosate [18,24]. At our Havatselet site, both of these methods would be challenging due to the presence of large ramets on the face of the cliff. Biological control agents that reduce survivorship are under consideration, such as the soft-scale insect Pulvinariella mesembryanthemi and the generalist pathogenic fungus Sclerotinia sclerotiorum [8]. Regardless of the control method used, rapid evaluation of the population size and structure would help any management program, as it would provide the information needed for the learning process in adaptive management [69]. If the post-data processing procedures could be better optimized, the drone technology has great potential to allow for rapid monitoring of Carpobrotus populations.

References 51, 52, 54: These websites should be cited in this way? 

Check further for references and citations in the text.

We have re-checked the requirements for PLOS ONE and made the needed adjustments to the References section. 

Add the role of anonymous reviewers and Editors in the acknowledgments.

Have a nice work.

We now acknowledge our anonymous reviewers and editor in the Acknowledgements section of our revised manuscript. 

Reviewer #2: 

General comments: I revised the manuscript entitled "Demographic analysis of Israeli Carpobrotus populations: management strategies and future directions". I found some merits in the manuscript which deals an interesting topic, providing demographic data using drones. Nevertheless, the whole paper should be extensively revised and rewritten.

We thank Reviewer #2 for his/her detailed suggestions. We have revised the manuscript text significantly to address these comments and believe it has improved the manuscript. Please see our detailed responses below.

Title:

The title should be revised and focused on the type of research: the use of images obtained by drone. The study has only been conducted in one population. “Management strategies and future directions” do not seem justified to include it in the title.

We have revised our title to: “Demographic analysis of an Israeli Carpobrotus population”.

Abstract:

Line 22 (L22) – delete (UAVs). L29 – Aizoaceae italics. L31 – confirm by demonstrate.

We have made the suggested adjustments to the text of the manuscript.

Introduction:

The introduction should be restructured. Information is there but is pretty messy.

Invasive species and the importance of coastal ecosystems (L66-72) could be reduced and could join with the problem of invasive species (L35-36). The description of Carpobrotus (L37-43), its impacts (L44-51) and the investigations (L56-59) should be rewritten and joined in a paragraph. Research on eradication and the effects of control actions should be mentioned. See below recent papers:

Chenot et al. (2018) Restor Ecol 26:106-113. Lazzaro et al. (2020) Biologia 75:199-208. Buisson et al. (2020), Applied Vegetation ScienceVolume 24, Issue 1. Braschi et al. (2021) Biodiversity and Conservation (2021) 30:497–518. Fos et al. (2021) Plant Biosystems. The part dedicated to "Structured population models" in Carpobrotus (L 52-59, L 62- 65) and "Structured population models" in dune ecosystems (L72-79) should be shortened and joined in one paragraph.

We have substantially revised our introduction to address all of these comments. Our new structure is:

Paragraph 1 – Coastal ecosystems in particular provide vital ecosystem services and are threatened by biological invasions

Paragraph 2 – Introduction to Carpobrotus and the harm it causes to coastal communities and ecosystems

Paragraph 3 – Research on eradication and control actions (including all the citations suggested by this reviewer)

Paragraph 4 – Structured population models (now shortened, as suggested by this reviewer)

Paragraph 5 – The use of drones to study the demography of difficult-to-access populations on coastal ecosystems.

Paragraph 6 – The particular questions we will address in this study.

L37 – Carpobrotus N. Br. (Aizoaceae); L38 – sensitive by frail; L40 – Correct botanic names C. edulis and C. acinaciformis.

We have made the suggested adjustments to the text of the manuscript.

L50 – Allelopathic properties of the Carpobrotus litter also inhibit emergence (Fos et al., 2021). 

We have added this mechanistic information, together with its appropriate reference. See lines 54-55, which now read: […] matter content and releasing allelochemicals that hinder seed germination, seedling emergence and root growth of some native plants [16-18].

L55 – can be rewritten: “management, managers, management” in the same sentence.

We have re-written this sentence and combined it with the previous sentence. See lines 63-65, which now read: Structured population models (e.g., matrix population models [29], integral projection models [30]) of invasive plants provide a tool for generating comprehensive fitness estimates and identifying sensitive vital rates (e.g. survival, growth) to target with management [29,31-35].

L59-62 – “Succulents are an …. and Opuntia spp (29)” delete the paragraph, the information is not relevant.

As a result of our revision of the Introduction section, this paragraph has been reformulated and shortened, and it now also includes Reviewer #1’s request of citing another relevant paper to the matter (Ecology, Ethology and Evolution Battisti & Fanelli in press). See lines 43-44, which now read: Coastal ecosystems are prone to biological invasions, most notably by Carpobrotus spp., Lampranthus spp., Opuntia spp., Malephora crocea and Mesembrianthemum cristallinum [6,7].

L86-87 delete “unmanned aerial vehicles” and “Integral Projection Models”.

L92 – confirm by demonstrate.

We have made the suggested adjustments to the text of the manuscript.

Methods:

The methods are written in a single section and they must organize in specific sections.

We have reorganized the Methods section and now it includes the following subsections: Study site and genus, Data collection, Image processing and data extraction, Demographic modelling and Integral Projection Modelling analysis.

Fig. 1 – Include scale.

We now include the figure scale.

L112-119 reduce paragraph, relevant information is only L115-116.

The use of the terms “plant” and “ramet” is confusing.

We have shortened this paragraph. We are now consistent throughout the manuscript, using the term ramet instead of ‘plant’ or ‘individual’. 

L160 4 m instead of 4m; L190 delete “Integral Projection Model”.

We have made the suggested adjustments to the text of the manuscript.

Results:

The results should include descriptive information from the experimental data, for example: total area occupied by Carpobrotus, mean plant area, maximum and minimum plant area, frequency by size of plant area, number of flowers and fruits per plant, etc. and variations from 2018 to 2019.

We have added a new figure giving population summary statistics (Figure 2), and text summary to the beginning of the Results section. See lines 231-232, which now read: Ramets ranged in size from -6.04 to 3.31 m2 (natural log scale), and flower counts were heavily skewed towards smaller values with a few very large, highly reproductive ramets (Fig 2).

L 235 why C. edulis ???

We have revised the phrase, and changed that particular portion of the Results to Carpobrotus spp., as all the listed species (Carpobrotus edulis (L.) N.E.Br., Carpobrotus acinaciformis (L.) L.Bolus, and Carpobrotus chilensis (Molina) N.E.Br.) were included in our analysis.

Discussion:

The discussion should be revised, shortened and rewritten.

We have extensively revised the Discussion section to address the comments of both reviewers, and in the process, we have also shortened it.

Include table and figure references in the discussion. 

We now include table and figure references in the Discussion section.

L273 – Aizoaceae italics.

We have made this suggested change across the entire manuscript. 

L279 – delete “Integral Projection Models”.

We have restructured our Discussion section and, in the process, removed this phrase.

L269-275 – This paragraph is better included in the introduction.

As there was some repetitiveness in this text within the Introduction section, we have deleted it.

L282-284 – Delete “Individual …… vertical cliff”.

We made this suggested deletion.

L289-291 – “Third, it is … for Carpobrotus” - this sentence is contradictory with the references on invasive behavior and growth of Carpobrotus (see Campoy et al 2018). 

We have clarified our Discussion section here. Specifically, we state that stable population growth is likely typical for older and/or less disturbed populations, while younger/more recently disturbed populations, or those at lower densities, are likely to exhibit positive population growth. Furthermore, we distinguish between growth of ramets and population growth rates. See lines 276-282, which now read: 

Third, it is possible that stable population growth rates are typical for older, less disturbed Carpobrotus populations. Rapid ramet and population growth rates may characterize earlier stages of invasion, and slow down occurs once a site becomes more saturated. Thus, newer or more recently disturbed populations are likely to exhibit different behavior from the Havatselet population. Finally, we note that our study is one year long, and that environmental conditions across time, especially in the context of climate change, which might increase or decrease the population growth rate.

L297-298 – No data is presented regarding it, see Result comments. 

We now reference Figure 4, which supports this conclusion.

L303-329 – This part should be rewritten by joining the 2 paragraphs. Recent eradication works and the effects of control actions should be included and discussed. See papers in introduction comments. It would be interesting to discuss the contributions of this work and the potential of the use of drones at least in the monitoring of regenerated areas. 

We have extensively re-written this part, and we added the suggested references, which we found very helpful. We discuss the potential use of drones for monitoring to allow for adaptive management and cite the Hockings et al 2006 paper, suggested by Reviewer #1.

For control actions, see lines 292-308, which now read:

Eradication of Carpobrotus is the objective of many coastal management programs. Our elasticity analysis shows the importance of targeting large ramets for removal and our methods show the possibility of using drones to rapidly monitor populations and allow for adaptive management [69]. A variety of tactics have successfully killed Carpobrotus and allowed for native biodiversity recovery. Manual removal is feasible for accessible populations, and it may or may not be optimal to follow this up with the removal of dead shoots and litter. Leaving litter increases the risk Carpobrotus germination from seed and slows recovery of native plants, but decreases the problem of erosion, which is an important issue in sites with steep slopes [13,23,25]. Effective mortality of Carpobrotus and rapid recovery of native vegetation has also been demonstrated with chemical control using Glyphosate [18,24]. At our Havatselet site, both of these methods would be challenging due to the presence of large ramets on the face of the cliff. Biological control agents that reduce survivorship are under consideration, such as the soft-scale insect Pulvinariella mesembryanthemi and the generalist pathogenic fungus Sclerotinia sclerotiorum [8]. Regardless of the control method used, rapid evaluation of the population size and structure would help any management program, as it would provide the information needed for the learning process in adaptive management [69]. If the post-data processing procedures could be better optimized, the drone technology has great potential to allow for rapid monitoring of Carpobrotus populations.

For the potential of the use of drones, see lines 315-329, which now read:

We were successful in using the drone for creating the maps of the Carpobrotus population and later use those maps for evaluating vital rates and parameterizing an IPM. This is another potential ecological use case for a rapidly developing technology used in both animal [53,74,75] and plant population studies [49,76,77]. While in our case, the drone successfully generated maps and subsequent estimates of ramet size, we acknowledge that they have their limitations. Georeferencing can prove challenging in environments without many permanent features to mark across multiple samples. If maps are not well aligned, re-identifying individuals from the previous year can be quite challenging. Advances in RTK and PPK GPS technology have greatly reduced this challenge, though these can still be quite pricey relative to typical conservation budgets. Furthermore, size estimates can vary greatly when the absolute distance between the ground and the camera varies across the orthomosaic scene, necessitating the placement of ground truth targets in areas that may be difficult to physically reach. Additionally, dense vegetation mats can confound the orthomosaic generating procedure. Even when it is successful, it may be difficult to identify individual plants. Therefore, we recommend that vegetation density and height be considered when choosing the mapping method. 

L305-306 – “It is … growth” too speculative. 

We removed the speculative sentence.

L310-311 – In the case of Carpobrotus, the actions should be aimed at the complete eradication and not at the reduction of the population. 

We agree. We have revised this part of the Discussion section substantially in response to this and the other comments about Carpobrotus management. See lines 292-308, which now read:

Eradication of Carpobrotus is the objective of many coastal management programs. Our elasticity analysis shows the importance of targeting large ramets for removal and our methods show the possibility of using drones to rapidly monitor populations and allow for adaptive management [69]. A variety of tactics have successfully killed Carpobrotus and allowed for native biodiversity recovery. Manual removal is feasible for accessible populations, and it may or may not be optimal to follow this up with the removal of dead shoots and litter. Leaving litter increases the risk Carpobrotus germination from seed and slows recovery of native plants, but decreases the problem of erosion, which is an important issue in sites with steep slopes [13,23,25]. Effective mortality of Carpobrotus and rapid recovery of native vegetation has also been demonstrated with chemical control using Glyphosate [18,24]. At our Havatselet site, both of these methods would be challenging due to the presence of large ramets on the face of the cliff. Biological control agents that reduce survivorship are under consideration, such as the soft-scale insect Pulvinariella mesembryanthemi and the generalist pathogenic fungus Sclerotinia sclerotiorum [8]. Regardless of the control method used, rapid evaluation of the population size and structure would help any management program, as it would provide the information needed for the learning process in adaptive management [69]. If the post-data processing procedures could be better optimized, the drone technology has great potential to allow for rapid monitoring of Carpobrotus populations.

L330-338 – This part is very general, and it has nothing to do with the work they present.

We reduced this part of the Discussion section. See lines 310-314, which now read:

To our knowledge, Israeli legislation does not include rules for the introduction and control of C. edulis and C. acinaciformis, and as such these are still allowed to be planted in urban gardens. This greatly hinders efforts for invasion prevention. Further demography studies could contribute to the identification of high-risk areas for which regulations could be put in place [8,54,73].

---

## [Decision Letter · Decision Letter 1]

16 Apr 2021

Demographic analysis of an Israeli Carpobrotus population

PONE-D-20-29490R1

Dear Dr. Bogdan,

We’re pleased to inform you that your manuscript has been judged scientifically suitable for publication and will be formally accepted for publication once it meets all outstanding technical requirements.

Kind regards,

Mirko Di Febbraro

Academic Editor

PLOS ONE

Additional Editor Comments (optional):

Reviewers' comments:

Reviewer's Responses to Questions

**Comments to the Author**

1. If the authors have adequately addressed your comments raised in a previous round of review and you feel that this manuscript is now acceptable for publication, you may indicate that here to bypass the “Comments to the Author” section, enter your conflict of interest statement in the “Confidential to Editor” section, and submit your "Accept" recommendation.

Reviewer #1: All comments have been addressed

Reviewer #2: All comments have been addressed

2. Is the manuscript technically sound, and do the data support the conclusions?

Reviewer #1: Yes

Reviewer #2: Yes

3. Has the statistical analysis been performed appropriately and rigorously? 

Reviewer #1: Yes

Reviewer #2: I Don't Know

4. Have the authors made all data underlying the findings in their manuscript fully available?

Reviewer #1: Yes

Reviewer #2: Yes

5. Is the manuscript presented in an intelligible fashion and written in standard English?

Reviewer #1: Yes

Reviewer #2: Yes

6. Review Comments to the Author

Reviewer #1: The authors followed all the reviewer's suggestions. Good. Check in references if the names of Jornlas are correctly reported (e.g. in row 407 I think that the appropriate name is Biologia (Bratislava), etc.). Good paper.

Reviewer #2: General comments

I revised the second version of the manuscript entitled “Demographic analysis of an Israeli Carpobrotus population”. I found the manuscript deeply ameliorated and the aspects and comments indicated in the first review have been considered and included.

I think it is worthy of being published in its current form.

7. PLOS authors have the option to publish the peer review history of their article (what does this mean?). If published, this will include your full peer review and any attached files.

Reviewer #1: No

Reviewer #2: No

---

## [Editor Report · Acceptance letter]

20 Apr 2021

PONE-D-20-29490R1 

Demographic analysis of an Israeli *Carpobrotus* population 

Dear Dr. Bogdan:

I'm pleased to inform you that your manuscript has been deemed suitable for publication in PLOS ONE. Congratulations! Your manuscript is now with our production department. 

Kind regards, 

on behalf of

Dr. Mirko Di Febbraro 

Academic Editor

PLOS ONE